# Variation in Antibiotic Treatment Failure Outcome Definitions in Randomised Trials and Observational Studies of Antibiotic Prescribing Strategies: A Systematic Review and Narrative Synthesis

**DOI:** 10.3390/antibiotics11050627

**Published:** 2022-05-06

**Authors:** Rebecca Neill, David Gillespie, Haroon Ahmed

**Affiliations:** 1Division of Population Medicine, Cardiff University, Cardiff CF14 4YS, UK; neillr@cardiff.ac.uk; 2Centre for Trials Research, Cardiff University, Cardiff CF14 4YS, UK; gillespied1@cardiff.ac.uk

**Keywords:** treatment failure, observational study, randomised trial

## Abstract

Antibiotic treatment failure is used as an outcome in randomised trials and observational studies of antibiotic treatment strategies and may comprise different events that indicate failure to achieve a desired clinical response. However, the lack of a universally recognised definition has led to considerable variation in the types of events included. We undertook a systematic review of published studies investigating antibiotic treatment strategies for common uncomplicated infections, aiming to describe variation in terminology and components of the antibiotic treatment failure outcomes. We searched Medline, Embase, and the Cochrane Central Register of Clinical trials for English language studies published between January 2010 and January 2021. The population of interest was ambulatory patients seen in primary care or outpatient settings with respiratory tract (RTI), urinary tract (UTI), or skin and soft tissue infection (SSTI), where different antibiotic prescribing strategies were compared, and the outcome was antibiotic treatment failure. We narratively summarised key features from eligible studies and used frequencies and proportions to describe terminology, components, and time periods used to ascertain antibiotic treatment failure outcomes. Database searches identified 2967 unique records, from which 36 studies met our inclusion criteria. This included 10 randomised controlled trials and 26 observational studies, with 20 studies of RTI, 12 of UTI, 4 of SSTI, and 2 of both RTI and SSTI. We identified three key components of treatment failure definitions: prescription changes, escalation of care, and change in clinical condition. Prescription changes were most popular in studies of UTI, while changes in clinical condition were most common in RTI and SSTI studies. We found substantial variation in the definition of antibiotic treatment failure in included studies, even amongst studies of the same infection subtype and study design. Considerable further work is needed to develop a standardised definition of antibiotic treatment failure in partnership with patients, clinicians, and relevant stakeholders.

## 1. Introduction

Antibiotic treatment failure has been described for decades, with reports even in the 1940s of treatment failure with streptomycin for typhoid fever and penicillin for syphilis [1,2]. There is no universally recognised definition of “antibiotic treatment failure”, but the term implies that the provision of antibiotics failed to achieve a desired clinical response. It is used as an outcome in randomised trials and observational studies of antibiotic treatment strategies and may comprise different events that could each indicate failure to achieve a desired clinical response. However, there is considerable variation in the types of events included in antibiotic treatment failure outcomes, even amongst studies with the same design, population, and setting.

For example, two double-blind randomized placebo-controlled trials comparing three days of oral amoxicillin with placebo for children aged 2 to 59 months with non-severe pneumonia in Pakistan used different criteria to ascertain their primary outcome of antibiotic treatment failure [3,4]. In the first trial [3], antibiotic treatment failure during the three-day intervention or control period was met if any of the following events occurred: death, development of WHO-defined danger signs, retraction of the lower chest wall, hospitalisation, new-onset infection, or a serious adverse event necessitating physician recommended change to the trial regimen. In the second [4] antibiotic treatment, failure was determined with only lower chest wall retraction and WHO-defined danger signs. Variation is more substantial amongst observational studies where retrospective designs and different data sources often dictate the use of different proxies of possible antibiotic treatment failure, ascertained over different time periods relative to the index event. Of several observational studies of different antibiotic treatment strategies for community acquired pneumonia in the USA, antibiotic treatment failure outcomes comprised events such as a subsequent outpatient consultation, emergency department visit, or hospital admission, with or without an antibiotic prescription or infection-related diagnostic code [5,6,7,8,9]. Outcome ascertainment started at 0, 1, or 3, days following the index event, and ascertainment periods varied between 7, 14, and 30 days. Similar differences exist amongst observational studies of antibiotic treatment for urinary tract infections, where treatment failure outcomes include any number of events, such as a record of a new prescription, an outpatient or emergency department visit, or a hospital admission [10,11,12].

Reducing variation in how studies define an antibiotic treatment failure outcome could make findings from different studies more comparable, thus reducing heterogeneity and facilitating a more meaningful synthesis of findings across studies. It could also increase the number of studies that include components of treatment failure that are important to patients, clinicians, and policymakers. The ultimate aim should be to co-develop a standardised definition of antibiotic treatment failure with relevant stakeholders that is easily adaptable to different infections and study designs. As a first step toward this aim, we systematically reviewed recently published randomised trials and observational studies that used an antibiotic treatment failure outcome to assess different antibiotic prescribing strategies for uncomplicated respiratory tract, urinary tract, and skin infections in ambulatory patients. Our objective was to describe variation in terminology and components of the treatment failure outcome and identify any patterns related to infection type or study design.

## 2. Results

Database searches identified 2967 unique records, of which 2127 were excluded based on title alone and 774 were excluded based on the abstract (Figure 1). We retrieved 65 studies for a full-text review, and 36 of those studies met our inclusion criteria (Table 1). This included 10 randomised controlled trials and 26 observational studies. Most studies were from the USA (n = 8), followed by the UK (n = 6) Taiwan (n = 5) and Pakistan (n = 3). Twenty studies exclusively investigated RTI, twelve investigated UTI, four investigated SSTI, and two investigated both RTI and SSTI.

### 2.1. Terminology

“Antibiotic treatment failure” was used in 25 (69.4%) studies, “clinical failure” in 5 (13.9%) studies, and “therapy failure” in 2 (5.6%) studies. The remaining four studies used “non-response”, “clinical treatment failure”, “treatment effectiveness”, and “failure response”. “Antibiotic treatment failure” was used frequently across infection subtypes and study designs. Of the five studies which use “clinical failure”, three were RCTs and two were retrospective cohort studies. “Therapy failure” was used in two studies, an SSTI and an RTI study, both of which were double-blind RCTs.

### 2.2. Components of Antibiotic Treatment Failure Outcomes

A total of 41 different components were used across the studies (Appendix A). The most common component in studies of RTI and UTI was an additional antibiotic prescription that either comprised a further course of the index antibiotic or a course of a different antibiotic. Hospitalisation and emergency department (ED) visit were common components across all infections and were used in 26 studies. Death was a common component in studies of RTI (n = 7 studies).

We found common themes during data extraction of components of treatment failure outcomes. Components could be broadly categorised as either a prescription change, change in clinical condition, or escalation of care. These were not predetermined categories but evolved as the review progressed. Of the 41 different components across the three infection subtypes, 17 related to prescription changes, 20 to changes in clinical condition, and 4 to escalation of care.

### 2.3. Respiratory Tract Infections

We identified 27 different components of treatment failure outcome definitions across 21 studies of RTI. Using the categories determined during data extraction, nine components related to prescription changes, fourteen to changes in clinical condition, and four to escalation of care. Five studies used components from a single category, seven from two categories, and six from all three categories.

Prescription changes were used as a component within 16 studies, of which three used prescription changes alone to define their treatment failure outcome. Change to a different antibiotic was the most common prescription change and was used in seven studies. The timeframe within which a prescription change could be used to ascertain the treatment failure outcome varied between studies; four studies used a timeframe of thirty days from the index antibiotic, two used fourteen days from the index prescription, and one (an RCT) considered treatment failure as a prescription change up to day four in the intervention group and day six in the control group.

Changes in clinical condition alone comprised the treatment failure outcome definition in five studies, and in combination with prescription change or escalation of care in eight studies. Of the five studies solely considering clinical condition in their definition, three investigated RTI treatment in paediatric patients and used development of WHO Integrated Management of Childhood Illness defined danger signs as a component of their definition. The most common clinical condition components were death and development of WHO danger signs. However, as with prescription change, the timeframe within which death or WHO danger signs could be used to ascertain the treatment failure outcome varied between studies (Figure 2 and Figure 3).

### 2.4. Urinary Tract Infections

We identified 12 different components of treatment failure outcome definitions across 12 studies of UTI: eight components related to prescription changes, two to changes in clinical condition, and two to escalation of care. Nine studies used components from a single category, and three from two categories.

Prescription changes were used as a component within 10 studies, of which seven used prescription changes alone to define their treatment failure outcome. Prescription changes broadly comprised new prescriptions of a different antibiotic, refill of index antibiotic, or both, with and without a clinical record suggesting ongoing symptoms. Only two studies used the same definition with them being by the same author. Differences in prescription changes include variations in the time used to attribute a prescription change to treatment failure, ranging from 14 to 42 days, and inclusion of additional conditions (Figure 4).

Escalation of care was used as a component within 10 studies [10,11,12] Only one used escalation of care of the sole definition component, with it being “either hospitalisation or emergency department visits for UTI”. Only one study used changes in clinical condition components, which they defined as “persistence or progression of any clinical UTI signs or symptoms or appearance of new signs or symptoms”.

### 2.5. Skin and Soft Tissue Infections

We identified 12 different components of treatment failure outcome definitions across five studies of SSTI: two components related to prescription changes, seven to changes in clinical condition, and three to escalation of care. Two used components from a single category, and three used components from all three categories, of which two were identical definitions.

Prescription change components were used within three studies. Of these, change in antibiotic was studied over varying time frames, with two studies choosing within 30 days and one study between 72 and 96 h.

Escalation of care factors were used within three study treatment failure definitions. The two studies with identical definitions considered hospitalisation, referral to specialist, or emergency department visit within 30 days to be indicative of treatment failure. The remaining study using an escalation of care factor used hospitalisation but considered within 72–96 h to be treatment failure.

Clinical condition factors within definitions varied in time frame, ranging from 72 h to 30 days, and outcome studied (Table 2).

## 3. Discussion

There is substantial variation in the definition of antibiotic treatment failure used in observational studies and randomised trials that extends to studies of the same infection subtype and the same study design. We identified three key components of treatment failure definitions: prescription changes, escalation of care, and clinical condition features. Prescription changes were most popular within UTI study definitions, while clinical condition features accounted for the majority of components in RTI and SSTI studies.

Despite substantial variation, there were some components common to studies of the same infection. In paediatric pneumonia studies, for example, WHO danger signs and hospitalisation were common components. Similarly, treatment change and hospitalisation were common within antibiotic treatment failure definitions in studies of community acquired pneumonia. However, despite similarities in the components used, there remained variation in the time frames used to ascertain components across all infection subtypes and study designs. For example, symptomatic changes in SSTI studies were studied over a range of 72 h to 21 days [20,29]. While variations in timeframe may be a necessity to account for differing pharmacokinetics of various antibiotics, these disparities may potentially indicate a lack of agreement on important indicators and features of antibiotic treatment failure. Additionally, different patient presentations, non-adherence, comorbidities, and different guidelines for first-line antibiotics further complicate the formation of a universal definition.

This was an extensive systematic review with a comprehensive search strategy, closely following the Cochrane methodology for systematic reviews. The risk of selection bias was reduced by an independent review of abstracts by two authors, with disputes resolved by a third. In addition, multiple databases were searched to maximise the likelihood of including all relevant papers.

Our study had some limitations. Our search strategy was limited to English language studies published since 2010. This may have resulted in additional relevant papers not being included in this review. There was limited opportunity to compare studies by antibiotic due to large variety of antibiotics studied within the papers included. The studies within this review were largely carried out within higher income countries, with the exception of studies on paediatric pneumonia treatment, as these tended to be carried out in lower-income countries. Studies were not assessed for risk of bias, given that the aim of this review was to describe components of antibiotic treatment failure outcomes and how they varied across studies, rather than assess individual and pooled effects of the interventions.

We categorised components of antibiotic treatment failure definitions iteratively—this was not prespecified and evolved as data extraction took place.

While this review highlights the variation in the definitions used across studies, we have not estimated the effect that this may have on treatment failure rates, due to the heterogeneity present even within studies of the same infection.

Numerous studies have investigated antibiotic treatment by using a treatment failure outcome, but little was known about what components this outcome comprised and how it was defined across studies. We found a lack of consensus definition within study designs and infection subtypes but found that components of treatment failure definitions could be split into prescription, escalation of care, and clinical condition components. These categories of components have been described previously [39] and partly align with proposed criteria to define treatment failure in hospitalised patients, using clinical, treatment, laboratory, and radiological parameters [40]. In the proposed criteria, clinical parameters refer to symptomatic changes and deterioration, and treatment parameters refer to additional antibiotics or other treatment, including admission to ITU (escalation of care). Microbial susceptibility and inflammatory markers, such as leucocytes, neutrophils, and CRP, are included within laboratory parameters, with the final criteria of radiological parameters covering the progression of radiological images. The use of radiological parameters is potentially less well suited to studies of community acquired infections and their management in primary care, due to the lack of readily available radiological imaging to practitioners.

Antibiotic prescribing is heavily influenced by national and regional guidelines. These guidelines are informed by research such as the studies included in this review, with observational studies providing an increasing amount of evidence. More antimicrobial stewardship work coming from observational studies makes a consensus definition for observational studies increasingly important.

Forming a consensus definition would have its challenges, requiring discussion and agreement from experts and the need to be specific for antibiotic or infection. However, a consensus definition has been reached for HIV treatment failure, with it being split into immunological and virological failure and based on fall in level of CD4+ cells and level of viral replication suppression, respectively, demonstrating the potential to have a treatment failure definition specific to each infection [41]. Reducing heterogeneity in study outcomes would aid comparing study results. This is important when developing antibiotic prescribing guidelines and facilitating antimicrobial stewardship.

## 4. Materials and Methods

This is a systematic review and narrative synthesis of published studies investigating antibiotic treatment strategies for common uncomplicated infections, with antibiotic treatment failure as either a primary or secondary outcome.

### 4.1. Search Strategy

The review was undertaken between September 2020 and May 2021. We searched Medline, Embase and the Cochrane Central Register of Clinical trials for English language studies published between January 2010 and November 2020. The search strategy comprised of a comprehensive list of medical subject heading (MeSH) terms for antibiotics, treatment failure, and the three common bacterial infections of interest: respiratory tract, urinary tract and skin and soft tissue infections. The search was initially run through Medline and then refined to make it compatible with the remaining databases. The Medline search is available in Appendix B.

### 4.2. Inclusion Criteria

We included observational studies or randomised controlled trials published in English between January 2010 and January 2021 if they met the following criteria:
(1)The population of interest was ambulatory patients seen in primary care or outpatient settings with respiratory tract (RTI), urinary tract (UTI), or skin and soft tissue infection (SSTI);(2)The comparisons were between antibiotic prescribing strategies—i.e., antibiotic versus antibiotic, or antibiotic versus placebo, or comparison of different doses or duratio ns of antibiotic therapy.(3)The outcome was antibiotic treatment failure. The exact terminology could vary, but the authors needed to describe an outcome that assessed failure to achieve a desired clinical response following initiation of antibiotics or control.

We excluded studies of non-bacterial infections, complicated infections (e.g., pyelonephritis), hospital inpatients, patients selected based on the presence of a specific co-morbidity (e.g., UTI in people who had experienced a stroke), and those where the antibiotic prescribing strategy followed a surgical procedure (which we regarded as complicated infections).

### 4.3. Screening and Data Extraction

RN ran the searches and excluded irrelevant studies based on their titles. RN and HA then independently screened titles and abstracts of the remaining articles for inclusion. Disagreements were resolved by discussion and arbitration by DG. RN retrieved full texts of potentially eligible studies. Eligibility was checked against the inclusion and exclusion criteria by HA and DG and discussed with RN to decide on the final list of included studies.

Data extracted from each study included design; country; year of publication; infection; details of interventions and controls; and detailed information about the terminology, description, and ascertainment of the antibiotic treatment failure outcome. Individual components of the antibiotic treatment failure outcomes were identified and extracted.

### 4.4. Synthesis

We narratively summarised key features from eligible studies by the type of infection. We used frequencies and proportions to describe terminology, components, and ascertainment periods of antibiotic treatment failure outcomes. We listed components of antibiotic treatment failure outcomes and then iteratively categorised them as patterns that emerged. We explored patterns by infection type and study design.

## 5. Conclusions and Implications

This review highlights the substantial variation in antibiotic treatment failure definitions across infection types and study designs. A consensus definition of treatment failure, with infection and antibiotic-specific features, would aid in comparing studies and their outcomes, and this, in turn, would help to guide more judicial prescribing of antibiotics. The work to form a consensus definition for antibiotic treatment failure has been started for urinary tract infections by the outline of a protocol for forming a core outcome set [42]. Future research to compare effect estimates for “antibiotic treatment failure” for the same infection by study design and outcome definition, while exploring how this, in turn, affects the conclusion of the antibiotic treatment being studied, would further determine the impact a lack of consensus definition may be having.

## Figures and Tables

**Figure 1 antibiotics-11-00627-f001:**
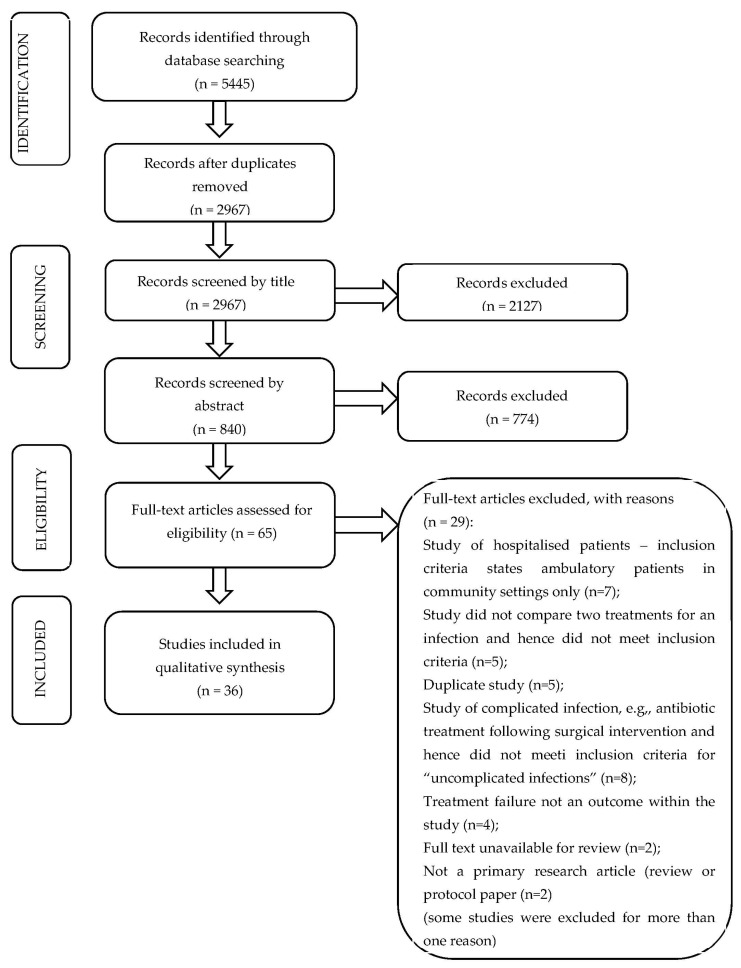
PRISMA diagram of included and excluded studies.

**Figure 2 antibiotics-11-00627-f002:**
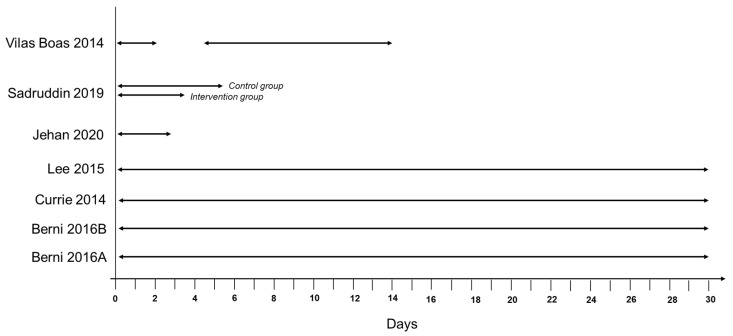
Time in days for ascertaining death in RTI studies.

**Figure 3 antibiotics-11-00627-f003:**
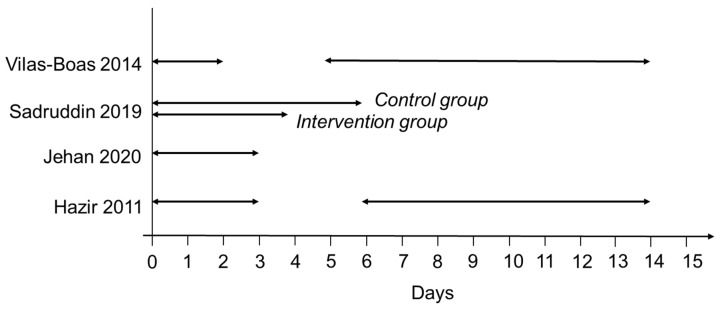
Time in days for ascertaining WHO danger signs in RTI studies.

**Figure 4 antibiotics-11-00627-f004:**
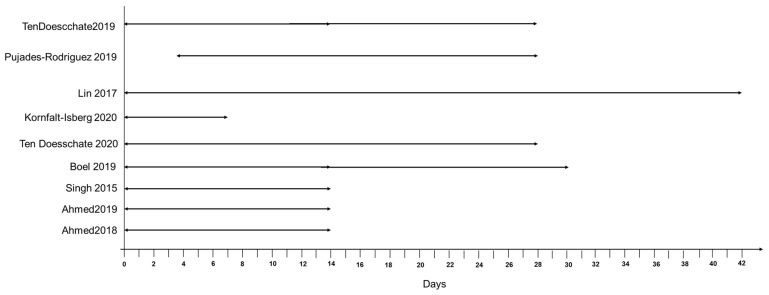
Time in days for ascertaining prescription change attributable to treatment failure in UTI studies.

**Table 1 antibiotics-11-00627-t001:** Summary of included studies.

Study ID and Country	Design	Sample Size	Infection	Intervention and Control	Antibiotic Treatment Failure Definition
Ahmed 2019A [13] UK	Retrospective cohort study	N = 33,745	UTI	3 days vs. 7 days of antibiotic therapy	Reconsultation for urinary symptoms and a same-day antibiotic prescription within 14 days following the incident UTI, ascertained through clinical and prescription codes recorded in primary care records.
Ahmed 2019B [14] UK	Retrospective cohort study	N = 42,298	UTI	Cefalexin, ciprofloxacin, or co-amoxiclav vs. Nitrofurantoin	Reconsultation for urinary symptoms and a same-day antibiotic prescription within 14 days following the incident UTI, ascertained through clinical and prescription codes recorded in primary care records.
Al-Saadi 2018 [15] Iraq	Retrospective cohort study	N = 120	Bacterial rhinosinusitis	Ceftriaxone 1 g IM once daily vs. amoxicillin + clavulanic acid 875 mg/125 mg bi-daily for 3–4 days	Persistence of signs and symptoms of acute bacterial rhinosinusitis or complications.
Ambroggio 2015 [5] USA	Retrospective cohort study	N = 1999 (1164 after matching)	CAP	Beta-lactam monotherapy vs. macrolide monotherapy	Follow-up visit with an ICD-9 code for a respiratory-related diagnosis accompanied by a change in antibiotic therapy either in the outpatient setting (in-person or via phone), in the emergency department, or as a hospital admission within 14 days of the initial diagnosis of CAP.
Ambroggio 2016 [6] USA	Retrospective cohort study	N = 915 (717 after restricted by propensity score)	CAP	Beta-lactam monotherapy vs. meta-lactam/macrolide therapy	Follow-up visit with an ICD-9 code for a respiratory-related diagnosis accompanied by a change in antibiotic therapy either in the outpatient setting (in-person or via phone), in the emergency department, or as a hospital admission within 14 days of the initial diagnosis of CAP.
Berni 2016A [16] UK	Retrospective cohort study	N = 7,471,893	RTIs	Wide range of different antibiotics were compared	Earliest occurrence of any of five events: 1. A different antibiotic dispensed between 1 and 30 days from the last prescription of the antibiotic monotherapy; 2. A GP record of hospitalization with an infection-related diagnostic code within 30 days of antibiotic initiation; 3. A GP referral to a specialist service within 30 days of antibiotic initiation, where the specialty type was infection-related, or the referral had an infection-related diagnostic code; 4. A GP record of an emergency department visit within 3 days of antibiotic initiation; 5. A GP record of death with an infection-related diagnostic code within 30 days of antibiotic initiation.
Berni 2016B [17] UK	Retrospective cohort study	N = 824,651	SSTI, and RTI	Wide range of different antibiotics were compared	Earliest occurrence of any of five events: 1. A different antibiotic dispensed between 1 and 30 days from the last prescription of the antibiotic monotherapy; 2. A GP record of hospitalization with an infection-related diagnostic code within 30 days of antibiotic initiation; 3. A GP referral to a specialist service within 30 days of antibiotic initiation, where the specialty type was infection-related, or the referral had an infection-related diagnostic code; 4. A GP record of an emergency department visit within 3 days of antibiotic initiation; 5. A GP record of death with an infection-related diagnostic code within 30 days of antibiotic initiation.
Blin 2010 [18] France	Prospective cohort study	N = 5640	Acute sinusitis	Antibiotic prescribed vs. no antibiotic prescribed	Sinus drainage or new antibiotic prescription (switch or initiation) within 10 days.
Boel 2019 [10] Denmark	Retrospective cohort study	N = 21,864	UTI	Pivmecillinam 3 days vs. 5 days vs. 7 days	Redemption of any new prescription of antibiotic exclusively for UTIs, redemption of other antibiotic for UTI with specified indication for UTI on the prescription. Or admission to hospital due to UTI within 14 and 30 days.
Currie 2014 [19] UK	Retrospective cohort study	SSTI N = 2,568,230 URTI N = 4,236,574 LRTI N= 3,148,947	SSTI, & RTI	Wide range of different antibiotics were compared	Earliest occurrence of any of the following: 1. Prescription of a different antibiotic drug within 30 days of the first-line antibiotic; 2. GP record of admission to hospital with an infection related diagnosis within 30 days of antibiotic initiation; GP referral to an infection related specialist service within 30 days of initiation; 3. GP record of an emergency department visit within three days of initiation (the shorter time window being selected here to increase the probability that the emergency event was related to the infection); 4. Or GP record of death with an infection related diagnostic code within 30 days of initiation.
Dalen 2018 [20] Canada	Double-blind non-inferiority RCT	N = 206,195	SSTI	Cephalexin 500 mg QDS vs. cefazolin 2 g IV daily + probenecid 1 g PO daily	Hospital admission, Change in antibiotics (not due to an adverse event), or persistent or worsening signs and symptoms of SSTI following at least 72 h of antibiotic therapy, assessed between 72 and 96 h after antibiotic therapy was initially started in the ED. Hospital admission or a change in antibiotics (not due to an adverse event) <72 h after the initial ED visit also constituted treatment failure.
Gerber 2017 [21] USA	Retrospective cohort study	N = 30,159 children (19,179 with acute otitis media; 6746, group A streptococcal pharyngitis; and 4234, acute sinusitis)	RTI	Broad spectrum vs. narrow-spectrum antibiotic therapy	Same acute infection diagnosis and a new prescription for a systemic antibiotic reported during an in-person or telephone encounter. If the encounter occurred during the effective duration of the initially prescribed (index) antibiotic course, progress notes were reviewed to distinguish between treatment failure (persistence of symptoms or concern for failure of the index antibiotic) and requirement of a new antibiotic due to an adverse event. Encounters with the same acute respiratory tract infection diagnosis and an antibiotic that occurred after the effective duration of the index antibiotic were considered treatment failures (i.e., recurrence) and did not require confirmatory record reviews. TF was assessed through 14 days as the primary outcome and 30 days after diagnosis. Assessment for TF started 2/7 after diagnosis.
Greenberg 2014 [22] Israel	Double-blind placebo-controlled RCT	N = 140 (12, 56 and 72 children in the 3-, 5- and 10-day treatment groups respectively)	CAP	2 stages: 3 days vs. 10 days; 5 days vs. 10 days amoxicillin (80 mg/kg/d; divided into 3 daily doses)	Judged by the study physicians to be nonresponsive or deteriorating to the point that the study drug needed to be replaced; or if the patient was hospitalized due to deterioration in medical condition or no response to the current treatment. Clinical relapse before day 30 was also defined as treatment failure. All arms received identical treatment for the first 3 days, so only failures after day 3 were included in the analysis.
Haghighi 2010 [23] Iraq	Double-blind RCT	N = 76	UTI	Ciprofloxacin 250 mg BD 3-days vs. ciprofloxacin 250 mg BD 7 days	Persistence or progression of any clinical UTI signs or symptoms or appearance of new signs or symptoms.
Hazir 2011 [4] Pakistan	Double-blind randomized placebo-controlled RCT	N = 873	Non severe pneumonia	Amoxicillin 45 mg/kg/day 3-days vs. placebo	Treatment failure by day 3 = developed lower chest indrawing or any of the general danger signs. Treatment failure days 6–14 = presence of fast breathing, lower chest indrawing, or general danger signs after clinical resolution on day 3.
Hess 2010 [7] USA	Retrospective cohort study	N = 3994	CAP	Wide range of different antibiotics were compared	≥1 of the following events: ≤30 days after index date: a refill for the index antibiotic after completed days of therapy, a different antibiotic dispensed >1 day after the index prescription; or hospitalization with a pneumonia diagnosis or emergency department visit >3 days post-index.
Huttner 2018 [24] Multi-country	RCT	N = 513	UTI	Nitrofurantoin 100 mg TDS 5-days vs. fosfomycin 3 g OD single dose	Need for additional or change in antibiotic treatment due to UTI or discontinuation due to lack of efficacy.
Jehan 2020 [3] Pakistan	Double-blind RCT	N = 4002	Fast breathing pneumonia	Amoxicillin vs. placebo	Any of the following: Death, WHO-defined danger sign, onset of lower chest indrawing; hospitalization for any reason; and change in study drug by study physician due to new onset comorbid infection or for serious non-fatal antibiotic associated adverse event. Assessed for on days 0, 1, 2, and 3 of randomisation in the morning and evening.
Kornfalt-Isberg 2020 [25] Sweden	Retrospective cohort study	N = 16,555	Lower UTI	Narrow spectrum vs. broad-spectrum antibiotic therapy	A new prescription of a different relevant UTI antibiotic within 7 days from index antibiotic prescription and a new registered lower UTI diagnosis.
Lee 2014 [12] Taiwan	Retrospective cohort study	N = 73,675	UTI	Wide range of different antibiotics were compared	Either hospitalization or emergency department visits for UTI.
Lee 2015 [26] Taiwan	Retrospective cohort study	N = 9256	CAP	Wide range of different antibiotics were compared	Composite of either one of the following events: second antibiotic prescription, hospitalization due to CAP, an emergency department visit with a diagnosis of CAP, or 30-day non-accident-related mortality.
LinC 2015 [27] Taiwan	Retrospective cohort study	N = 2622 matched-pair episodes	CAP	Fluoroquinolones vs. beta lactams	≥1 of the following events: prolonged antibiotic use of 14 days or more, a second antibiotic added from a different class other than the index drug, and a change from oral antibiotics to injected medication.
LinKY 2015 [28] Taiwan	Retrospective cohort study	N = 2592	CAP	Fluoroquinolones vs. β-lactam/β-lactamase inhibitors	Prolonged antibiotic treatment for more than 14 days; change or addition of another antibiotic different from study medication; switch from oral antibiotics to intravenous antibiotics.
Lin 2017 [11] Taiwan	Retrospective cohort study	N = 2434	UTI	Generic vs. branded antibiotic formulations	An ER visit or hospitalization due to a UTI with antibiotic prescription within 42 days of the index consultation; and an additional outpatient visit for a UTI requiring antibiotic treatment within 42 days of the completion of the original antibiotic therapy.
Llop 2017 [8] USA	Retrospective cohort study	N = 441,820 outpatients	CAP	Any oral fluroquinolone, macrolide, or beta-lactam monotherapy	30-day rate of treatment switch: Switch in drug class after the index window, accompanied by a second CAP diagnosis within the range of 3 days prior to and 5 days following the switch), and the rate of CAP-related hospitalizations (patients with a primary diagnosis of CAP when admitted to hospital) in the 30 days following the initiation of outpatient treatment.
Moran 2017 [29] USA	RCT	N = 500	SSTI	Cephalexin + trimethoprim-sulfamethoxazole vs. cephalexin + placebo	Fever; increase in erythema (>25%), swelling, or tenderness (days 3–4); no decrease in erythema, swelling, or tenderness (days 8–10); and more than minimal erythema, swelling, or tenderness (days 14–21).
Pujades-Rodriguez 2019 [30] UK	Retrospective cohort study	N = 494,675 UTIs (300,354 patients)	UTI	Trimethoprim, co-amoxiclav, pivmecillinam, or nitrofurantoin	Antibiotic re-prescription—earliest prescription of a UTI-specific antibiotic for the same UTI episode between 4 and 28 days after the date of the initial antibiotic prescription.
Rajesh 2013 [31] India	RCT	N = 240	CAP	Amoxicillin 40 mg/kg/day in vs. co-trimoxazole 8 mg/kg/day or trimethoprim	Occurrence of any signs of WHO-defined severe pneumonia; increase in respiratory rate more than 10 breaths per min above base line and respiratory rate more than 70 per min for children 2 months to 1 year of age or more than 60 per min for children between 1 year and 5 years of age. Assessed after 2 and 5 days.
Sadruddin 2019 [32] Pakistan	Unblinded cluster RCT	N = 15,749	Fast-breathing pneumonia	Amoxicillin vs. co-trimoxazole	Death; appearance of any danger sign (unable to drink/breastfeed, convulsions, vomits everything, abnormally sleepy/difficult to wake) up to day 4 in intervention-cluster patients or day 6 in control-cluster patients; appearance of lower chest indrawing anytime up to day 4 or 6; change of antibiotic (through self-referral or by caregivers) anytime up to day 4 in intervention-cluster patients or day 6 in control-cluster patients; or fast breathing (respiratory rate ≥50 breaths per minute) on day 4 in intervention clusters or day 6 in control clusters.
Singh 2015 [33] Canada	Retrospective cohort study	N = 191,857	UTI		14 days following prescription of an antibiotic: receipt of a second antibiotic indicated for urinary tract infection; and hospital presentation (either an emergency department visit or hospital admission) with a urinary tract infection.
Ten Doesschate 2019 [34] The Netherlands	Retrospective cohort study	N = 58,709 episodes in 36,439 patients	UTI	Nitrofurantoin 5 days, fosfomycin 1 day, and trimethoprim 3 days or 7 days	New antibiotic prescription for cystitis or pyelonephritis, combined with an ICPC code for urinary tract infections.
Ten Doesschate 2020 [35] The Netherlands	Retrospective cohort study	N = 42,473 episodes in 21,891 patients	UTI	Nitrofurantoin 5 days, Fosfomycin 1 day and Trimethoprim 3 days or 7 days	Second antibiotic prescription for cystitis or pyelonephritis within 28 days post-prescription.
Tillotson 2020 [9] USA	Retrospective cohort study	N = 251,947	CAP	Fluoroquinolone, macrolides, beta-lactam, or tetracycline	≥1 of the following events within 30 days of initial antibiotic: Antibiotic refill, antibiotic switch, emergency-room visit, or hospitalization.
Vandepitte 2015 [36] Thailand	Prospective observational study	N = 209 enrolled	URTI	Wide range of different antibiotics were compared to no antibiotics	No clinical improvement, or worsening.
Vilas-Boas 2014 [37] Brazil	RCT (triple-blinded and single centre)	N = 820	Non-severe pneumonia	Amoxicillin twice daily vs. three times daily	Development of danger signs, persistence of fever, tachypnoea, development of serious adverse reactions, death, and withdrawal from the trial. Recurrence of fever and previously defined as TF included for secondary outcome TF definition. Failure rate measured at 48 h and 5–14 days after enrolment.
Williams 2011 [38] USA	Retrospective cohort study	N = 41,094	SSTI	Clindamycin, trimethoprim-sulfamethoxazole, and β-lactam	An SSTI within 14 days after the incident SSTI.

**Table 2 antibiotics-11-00627-t002:** Clinical condition definition components used in SSTI studies.

Study ID	Clinical Condition Definition Components
Berni 2016	Death with infection related diagnostic code within 30 days
Currie 2014	Death with infection related diagnostic code within 30 days
Dalen 2018	Persistent/worsening symptoms within 72–96 h after starting treatment
Moran 2017	Fever; increase in erythema (>25%), swelling, or tenderness (days 3–4); no decrease in erythema, swelling, or tenderness (days 8–10); and more than minimal erythema, swelling, or tenderness (days 14–21)
Williams 2011	An SSTI within 14 days after index SSTI

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
