# Peer review of "Variation in Antibiotic Treatment Failure Outcome Definitions in Randomised Trials and Observational Studies of Antibiotic Prescribing Strategies: A Systematic Review and Narrative Synthesis"

_antibiotics, 2022, doi:10.3390/antibiotics11050627_

Round 1

Reviewer 1 Report

Authors investigated variations in antibiotic treatment failure outcome definitions. Paper is interesting, however, needs further work:

  • introduction could be expanded with additional information, as only two studies from Pakistan were emphasized; in such way, general message, literature gap and novelty of this investigation must be explored better
  • When was this review conducted?
  • What are detailed reasons for all excluded papers from this review?
  • Quality of assessed studies and risk of bias control should be expanded and mechanisms better emphasized. Furthermore, risk of bias could be more discussed and explored in Discussion section.
  • In discussion, limitations of this study should be added

Author Response

Authors investigated variations in antibiotic treatment failure outcome definitions. Paper is interesting, however, needs further work:

introduction could be expanded with additional information, as only two studies from Pakistan were emphasized; in such way, general message, literature gap and novelty of this investigation must be explored better

RESPONSE: Thank you for your comments on our manuscript. In response to your review, we have expanded and re-structured the introduction to include background information from more studies and to make the general message clearer.

When was this review conducted?

RESPONSE: The review was conducted between September 2020 and May 2021. We have added this to the Methods.

What are detailed reasons for all excluded papers from this review?

RESPONSE: We have added detail to the PRISMA flow chart (figure 1) for reasons for excluding papers

Quality of assessed studies and risk of bias control should be expanded and mechanisms better emphasized. Furthermore, risk of bias could be more discussed and explored in Discussion section.

RESPONSE: Studies were not assessed for risk of bias, given that the aim of this review was to describe components of antibiotic treatment failure outcomes and how they varied across studies, rather than assess individual and pooled effects of the interventions. We have highlighted this in the Discussion.

In discussion, limitations of this study should be added

RESPONSE: The Limitations paragraph has been highlighted in the discussion.

Reviewer 2 Report

The study is well documented and reasoned. I suggest following the instructions for the authors regarding the references.

Author Response

The study is well documented and reasoned. I suggest following the instructions for the authors regarding the references.

RESPONSE: Thank You for your comment and review. We have followed the author instructions for referencing.

Reviewer 3 Report

The Clinical issue of antibiotic treatment failure or antibiotic rational use has been an interesting topic in within the clinical field and with the scientific community.

This review highlights and emphasizes the substantial variation in antibiotic treatment failure definitions across infection types and study designs. Addressing the issue does not only tell us that the issue remains but also that we need to continue to advocate for monitoring and rational use.

The topic is of great interest and we thank the authors to point it out and to continue to raise awareness about its consequences.

Author Response

The Clinical issue of antibiotic treatment failure or antibiotic rational use has been an interesting topic in within the clinical field and with the scientific community.

This review highlights and emphasizes the substantial variation in antibiotic treatment failure definitions across infection types and study designs. Addressing the issue does not only tell us that the issue remains but also that we need to continue to advocate for monitoring and rational use.

The topic is of great interest and we thank the authors to point it out and to continue to raise awareness about its consequences.

RESPONSE: Thank you for your review and your positive comments.

Reviewer 4 Report

1. Line 40-61 is rather a discussion than an introduction. I suggest authors to restructure those lines.

2. It would be helpful to see the random effects analysis for the 36 studies included.

Author Response

Line 40-61 is rather a discussion than an introduction. I suggest authors to restructure those lines.

RESPONSE: Thank you for your review and your comment. Lines 40-61 have been re-structured but not removed as other reviewers requested they be expanded.

It would be helpful to see the random effects analysis for the 36 studies included.

RESPONSE: Thank you. The aim of this review was to use a narrative method to describe components of antibiotic treatment failure outcomes and how they varied across studies, rather than assess heterogeneity or effects quantitatively. We believe the narrative summary gives a more meaningful overview of the different components used to ascertain antibiotic treatment failure.

Round 2

Reviewer 1 Report

Authors have responded to all of the comments, and improved the manuscript. I have no further questions.